# Image polaritons in boron nitride for extreme polariton confinement with low losses

In-Ho Lee[1], Mingze He [2], Xi Zhang[3], Yujie Luo[3], Song Liu[4], James H. Edgar [4], Ke Wang[3], Phaedon Avouris[5], Tony Low[1], Joshua D. Caldwell [2] & Sang-Hyun Oh [1✉]

Polaritons in two-dimensional materials provide extreme light confinement that is difficult to achieve with metal plasmonics. However, such tight confinement inevitably increases optical losses through various damping channels. Here we demonstrate that hyperbolic phonon polaritons in hexagonal boron nitride can overcome this fundamental trade-off. Among two observed polariton modes, featuring a symmetric and antisymmetric charge distribution, the latter exhibits lower optical losses and tighter polariton confinement. Far-field excitation and detection of this high-momenta mode become possible with our resonator design that can boost the coupling efficiency via virtual polariton modes with image charges that we dub 'image polaritons'. Using these image polaritons, we experimentally observe a record-high effective index of up to 132 and quality factors as high as 501. Further, our phenomenological theory suggests an important role of hyperbolic surface scattering in the damping process of hyperbolic phonon polaritons.

[1] Department of Electrical and Computer Engineering, University of Minnesota, Minneapolis, MN, USA. [2] Department of Mechanical Engineering, Vanderbilt University, Nashville, TN, USA. [3] School of Physics and Astronomy, University of Minnesota, Minneapolis, MN, USA. [4] Tim Taylor Chemical Engineering Department, Kansas State University, Manhattan, KS, USA. [5] IBM T. J. Watson Research Center, Yorktown Heights, NY, USA. ✉email: sang@umn.edu

Surface phonon polaritons (SPhPs)[1–3] are collective oscillations of atomic lattice vibrations coupled with electromagnetic waves, supported by polar materials such as hexagonal boron nitride (hBN)[4,5], silicon carbide (SiC)[6], and molybdenum trioxide (MoO$_3$)[7]. These modes are supported within spectral regions bounded by the frequencies of the longitudinal (LO) $\omega_{LO}$ and transverse optic (TO) $\omega_{TO}$ phonon modes, which is referred to as the Reststrahlen band. This band is characterized by a negative real part of the permittivity. A characteristic feature of SPhPs[1,8,9] is their ability to confine radiation to subdiffractional length scales[10,11], benefitting various applications such as surface-enhanced infrared spectroscopy[4,12,13], thermal energy harvesting[14], and nanoscale heat transfer[3], among others. In contrast to diffraction-limited electromagnetic modes in dielectric waveguides, the polariton wavelength can be continuously reduced by shrinking the dimensions of polariton-supporting materials. In this regard, two-dimensional (2D) materials offer an exciting opportunity to explore highly confined polaritons with unprecedentedly large in-plane momenta[10,15–17]. Recent works on SPhPs in 2D materials[2] such as hBN and MoO$_3$ have reported effective indices, $n_{eff}$ ($= k/k_0$, where $k$ is the in-plane polariton wavevector and $k_0$ is the free-space wavevector) of around a hundred[10,18]—several times larger than values previously reported for plasmons in metallic nanostructures and comparable to graphene[19]. While graphene plasmons can be more tightly confined than metal plasmons[20], their confinement (and hence $n_{eff}$) is ultimately limited by Landau damping[21–24], through which a plasmon creates an electron-hole pair as it enters the single-particle phase space[25]. In contrast, SPhPs in hBN are unimpeded by Landau damping due to the bosonic nature of both phonons and photons, suggesting that polariton confinement via SPhPs in hBN can exceed what is possible with graphene plasmons.

Despite its distinctive physical origin, the SPhPs are phenomenologically similar to surface plasmon polaritons (SPPs)[26–28], which originate from collective oscillations of electrons. While strong confinement of polaritons can be achieved for SPP and SPhP modes near the plasma frequency $\omega_p$ and $\omega_{LO}$, respectively, extreme confinement has recently been demonstrated for graphene plasmons, when placed in proximity to a metal plate with a thin dielectric spacer. The modified Coulomb interactions lead to a linear dispersion known as an acoustic plasmon[29–36]. The acoustic graphene plasmons originate from coupling between the plasmons supported by a graphene layer and its mirror image inside the conducting plate with out-of-phase charge oscillations. As a result, most of the electric field is confined within the gap between graphene and the conducting plate, resulting in extreme plasmon confinement several times tighter than conventional graphene plasmons. The ultimate confinement limit of the acoustic graphene plasmons, however, is still set by Landau damping, and the maximum $n_{eff}$ is given by $c/v_F = 300$, with $c$ and $v_F$ being the speed of light and the graphene Fermi velocity[35]. Thus, the bosonic nature of the SPhPs raises intriguing questions regarding the fundamental characteristics of such image polaritons and their ultimate limit when Landau damping is no longer present.

SPhPs in hBN are long-lived[4,37], and can be harnessed to store electromagnetic energy in resonators[10,12,38] and propagating modes[37,39,40]. Since axial and tangential permittivities are of opposite signs in the two Reststrahlen bands of hBN (upper band $\lambda \approx 6.2$–$7.3\,\mu m$; lower band $\lambda \approx 12.1$–$13.2\,\mu m$), SPhPs in hBN feature a characteristic hyperbolic dispersion and thus are known as hyperbolic phonon polaritons (HPhPs)[10,16,41–45]. For propagating HPhPs, the figure-of-merit (FoM) can be defined as the ratio of the real part of the momentum to the imaginary part. FoM values of up to 33 have been measured by scattering-type scanning near-field optical microscopy[16]. For HPhP resonators, quality ($Q$) factors of 70 and 283 have been reported for the resonators based on hBN ribbons[12] and nanocones[10], respectively, which are one or two orders of magnitude larger than the $Q$ of graphene plasmonic resonators[36,46]. Recently, ultrapure, isotopically enriched hBN with either $^{10}$B or $^{11}$B (with longer phonon lifetimes than hBN with the natural 20% $^{10}$B and 80% $^{11}$B isotope distribution) has been utilized to further push the FoM to beyond 40 for propagating modes[37]. While these performance metrics are already impressive, their ultimate limit has yet to be experimentally probed and understood. Importantly, extreme subwavelength confinement via hBN implies that free-space photons cannot excite these high-momenta HPhP modes efficiently. To construct practical polaritonic devices it is desirable to simultaneously accomplish tight field confinement, highly efficient far-field coupling of light, and high $Q$-factors unimpeded by Landau damping.

Here, we solve these challenges by using monoisotopic h$^{10}$BN and our image polariton resonator with a high coupling efficiency to harness hyperbolic image phonon polariton (HiPP) modes. The resonator consists of an array of metal ribbon antennas located at a few nanometers below an hBN slab, which provides the momentum necessary to launch the hBN HPhPs. As in the acoustic plasmon case, the presence of image charges improves the confinement of the HPhPs, illustrating that such strategies can be universally applied for polaritonic modes. This high confinement was correlated with higher-order antisymmetric modes exhibiting record-high effective indices up to $n_{eff} = 132$ and $Q$ as high as 501, respectively. The symmetric modes are found to have relatively lower $Q$'s and smaller $n_{eff}$ compared to the antisymmetric counterpart, but exhibit higher coupling efficiencies allowing for the extinction of nearly 90% of incident light. Based on the strong dependence of the $Q$ upon the dielectric gap thickness, we suggest surface scattering effects as one of the primary damping pathways and develop the phenomenological theory for describing the loss mechanisms of the HiPPs. The high $Q$'s achieved for such large $n_{eff}$ are unprecedented compared with other electromagnetic modes such as SPPs in metals and graphene, where the increase in the confinement is compensated by increases in the optical loss.

## Results

**Image polaritons: concept and dispersion characteristics.** In the presence of a mirror plane within tens of nanometers from a polariton-supporting material, a polariton mode is mirrored, but with an inverted charge distribution (Fig. 1a). This strategy has been utilized to enhance the confinement of SPPs in noble metals via gap plasmons[47,48]. In the case of hBN, HPhP modes supported in a slab of thickness $t$ are quantized with the out-of-plane wavevector of $k_y = l\pi/t$, where $l$ denotes the mode number. For very thin films, coupling efficiencies to modes with large $l$ become negligible due to the large momentum mismatch, thus we consider the case where only two modes with $l = 1$ and 2 are practically accessible. The 1st mode exhibits polaritonic fields along the in-plane direction that oscillate in-phase and is referred to as the "symmetric mode" (Fig. 1b), while the 2nd mode features out-of-phase charge oscillations and is referred to as the "anti-symmetric mode" (Fig. 1c) (see Supplementary Fig. 1). Due to the out-of-phase charge oscillations, most of the electric fields for the antisymmetric mode are confined within the hBN slab, leading to stronger polariton confinement than for the symmetric mode. Due to the opposing charge distribution between the original and mirror polaritons, the electric fields are thus, more tightly confined within a dielectric gap between the hBN layer and conducting plane. The presence of a conducting plane can better

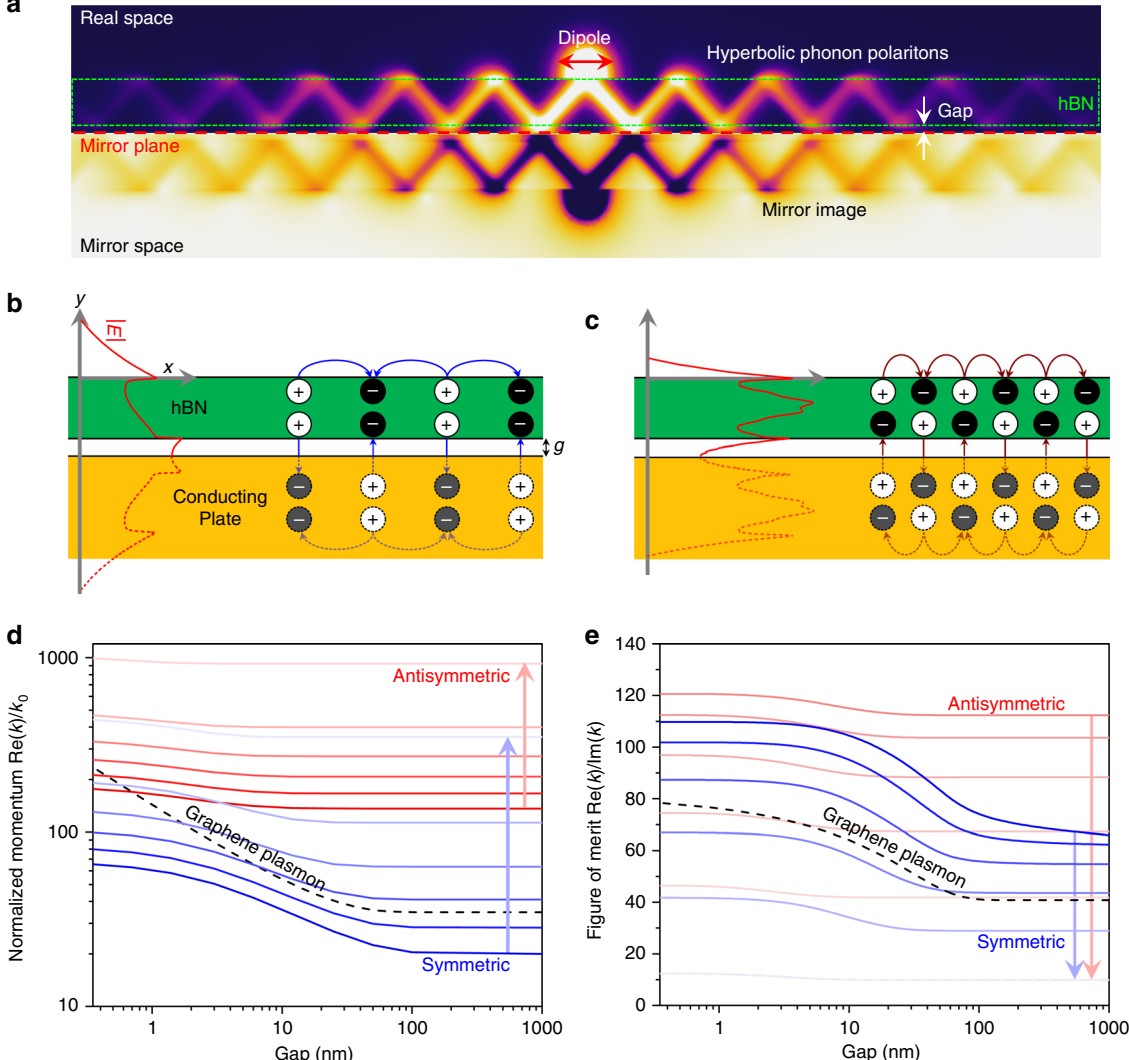

**Fig. 1 Concept of image polaritons. a** Schematic illustrations of image polaritons. Charge distributions for **b** symmetric mode and **c** anti-symmetric mode. In the presence of a mirror, image polaritons with out-of-phase charge oscillations are launched and confine radiation to the gap between the hBN layer and the mirror. **d** The normalized in-plane momentum and **e** figure of merit as a function of the gap size for the symmetric and anti-symmetric polariton modes, and graphene plasmons at five different frequencies of 1522, 1545, 1569, 1592, 1616, and 1639 cm$^{-1}$. The arrows represent the direction of increasing frequency. For graphene, the doping level and damping rate are 0.4 eV and 0.00378 fs$^{-1}$, respectively.

confine the in-plane HPhPs of different modal natures with the confinement increasing with decreasing dielectric gap size (Fig. 1d) (see supplementary note 1). Also, the in-plane momentum increases as the frequency of the incident wave approaches $\omega_{LO}$ in hBN. While in graphene the confinement measured by $n_{eff}$ is limited to $n_{eff} < c/v_F \sim 300$ due to Landau damping, and the maximum quality factor by electron–phonon scattering[46,49], the confinement in HiPPs is limited only by the intrinsic material loss. These HiPPs exhibit enhanced FoMs as the gap size scales down to the nanometer regime, and can also surpass that of its SPP analog in graphene (Fig. 1e). Here, we realize a highly efficient polaritonic nanoresonator with iso-topically enriched hBN to uncover the experimental upper bounds of these HiPPs, akin to the acoustic polaritons in graphene-based devices.

**Design and fabrication of image polariton resonators.** Far-field excitation and observation of image polaritons becomes possible with our resonator design illustrated in Fig. 2a. In contrast to previous resonator designs based on patterned hBN structures,

our approach utilizes an unpatterned hBN flake, minimizing the scattering of the polaritons at the patterned edges of resonators and eliminating additional damage induced through traditional lithographic patterning. Instead, the continuous conducting plane that was previously used for mirroring the polaritons as in Fig. 1b, c is patterned into an array of optical nanoribbon antennas that serve to both enable the mirror polariton, while also providing the necessary momenta to efficiently launch the polaritons. The scattered fields at the edges of the nanoribbons acquire high momenta and can excite image polaritons in both the metal-coupled hBN region and normal HPhPs in the metal-free region. The normal HPhPs efficiently convert into higher-momenta image polaritons due to the continuity of the hBN slab, and this two-stage coupling mechanism boosts the excitation efficiency compared to resonators based on patterned hBN nanoribbons[36]. The substrate that includes the metal ribbons, optical spacer, and reflector is fabricated using a template-stripping process[36,50,51], which produces an ultraflat top surface even in the presence of the undulating topography of the metal ribbons. The optical spacer and the reflector further boost the coupling efficiency by

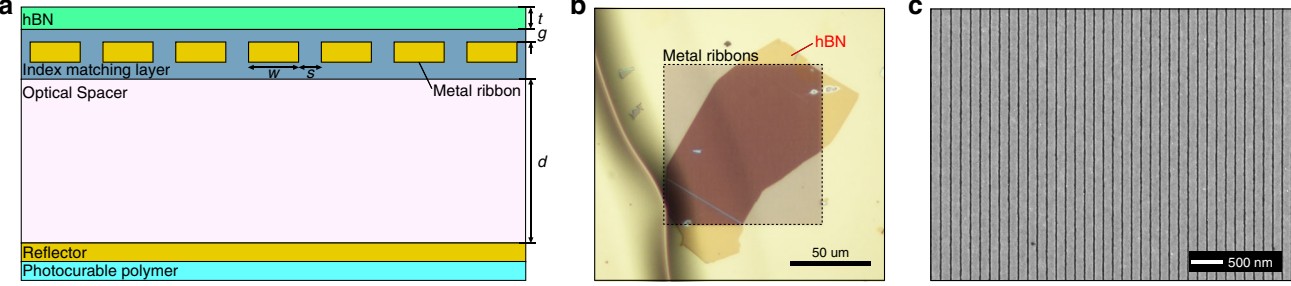

**Fig. 2 Device architecture. a** Schematic illustration of the image polariton resonator. $w$, $s$, $g$, and $t$ denote the width of a metal ribbon, the spacing between neighboring metal ribbons, the gap size between the hBN and the conducting plate, and the thickness of the hBN. $d$ denotes the thickness of an optical spacer, which is determined by the quarter wavelength condition to enhance the coupling efficiency of far-field light into image polaritons. **b** Optical microscope image of a h[10]BN flake transferred on the template-stripped substrate. **c** Scanning electron microscopy image of metal ribbons with $p = 150$ nm.

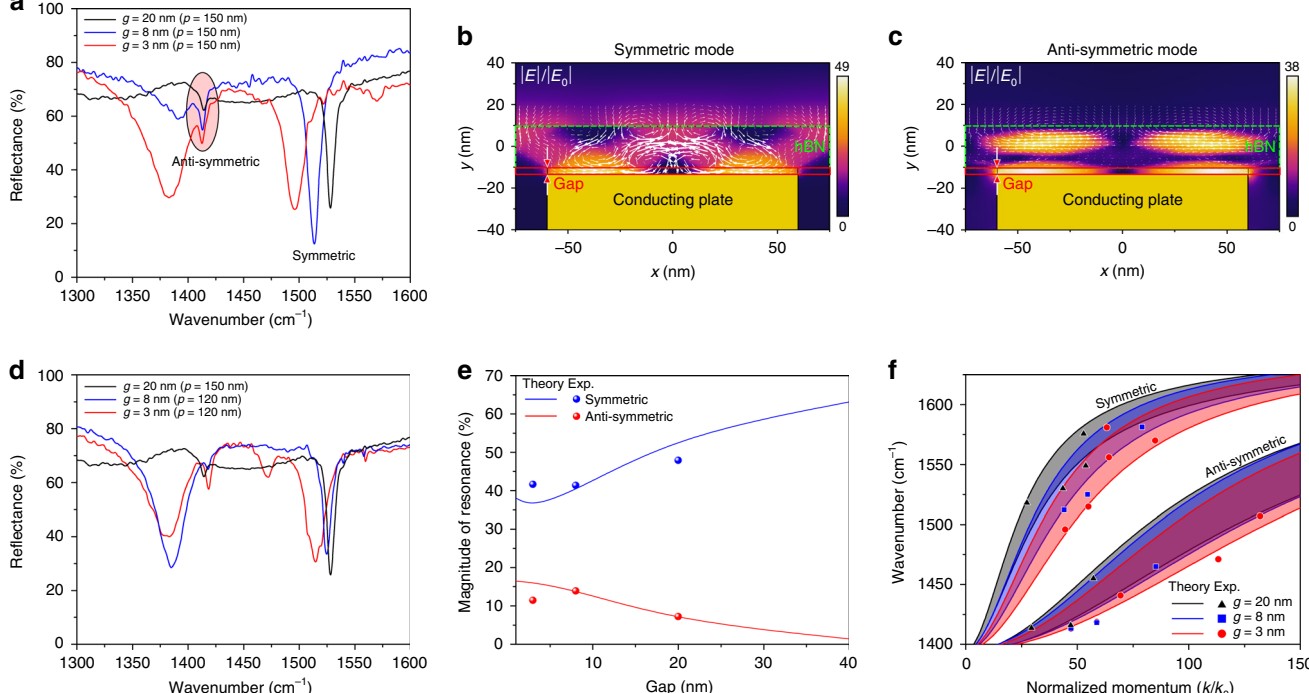

**Fig. 3 Gap dependence and dispersion for h[10]BN. a** The reflection spectra for $g = 3$, 8, and 20 nm given the periodicity of 150 nm. The spatial distribution of the electric fields for **b** the symmetric mode and **c** antisymmetric mode. The white arrows represent Poynting vectors. **d** Reflection spectra measured from the devices whose resonance wavelengths for the symmetric mode are almost aligned at the wavenumber of 1522 cm$^{-1}$. **e** The gap dependence of the magnitude of a resonance as a function of the gap sizes for the symmetric and anti-symmetric mode at around the frequency of 1522 and 1416 cm$^{-1}$, respectively. The solid lines denote the numerical results with constant scaling factors of 0.75 and 0.2 for the symmetric and antisymmetric case. **f** The in-plane momenta extracted from the measured spectra for different gap sizes of $g = 3$ (red circle), 8 (blue triangle), and 20 nm (black square) together with the analytical dispersions. For each gap size, the dispersion is calculated for $t = 20$ and 30 nm (the shaded area between two solid lines) to account for hBN thickness ($t$) variation in the fabricated samples.

recycling the transmitted waves back to the image polaritons when the thickness of the optical spacer is designed to satisfy the quarter-wavelength condition[36]. An optical micrograph in Fig. 2b shows the top view of the resonator with the atomically smooth hBN layer on top of metal ribbon patterns featuring varying dielectric gaps (see Supplementary Fig. 2 for topography). An electron micrograph in Fig. 2c shows that the top surface of the metal ribbons ($p = 150$ nm) is smooth and free of defects over the area of interest.

**Far-field excitation and detection of image polaritons.** The gap dependence of the image polariton dispersion has been characterized using far-field Fourier transform infrared (FTIR)

spectroscopy. The reflection spectra measured from devices with three different gap sizes of 3, 8, and 20 nm given the metal ribbon periodicity ($p$) of 150 nm are shown in Fig. 3a. The large resonance dips at around 1500 cm$^{-1}$ originate from the symmetric mode. Particularly for $g = 8$ nm, the resonant absorption is nearly 90%, showing the high coupling efficiency of the image polariton resonator enabled by the two-stage coupling mechanism and the photon-recycling scheme[36]. As the gap size decreases, the resonance shifts to lower frequencies at a given in-plane momentum. On the other hand, the smaller amplitude resonances near 1400 cm$^{-1}$ correspond to the antisymmetric modes, which are less sensitive to the gap size. The spatial distribution of electric fields for the symmetric mode in Fig. 3b shows characteristic hyperbolic

rays inside the hBN layer and highly confined fields within the gap region. For the antisymmetric mode (Fig. 3c), the field intensities inside the hBN layer and the gap are of similar magnitudes. The larger intensities inside the hBN layer are associated with the out-of-phase oscillations of polaritonic fields on the top and bottom surface of the hBN layer. Such spatial distributions also explain why the antisymmetric mode is less sensitive to the gap size. To investigate the gap-dependence of the resonant absorption, the far-field spectra from the resonators with similar resonance frequencies are compared (Fig. 3d). Instead of resonant absorption, the magnitudes of the Lorentzian fits for the peaks are extracted from the resonances for fair comparison (Fig. 3e). For the symmetric case, the resonances at around $1522 \, \text{cm}^{-1}$ are considered, while for the antisymmetric case, the resonances near $1416 \, \text{cm}^{-1}$ are selected. Due to the different modal natures of the two modes, the resonant absorption scales differently with the gap size. For the antisymmetric mode, placing the metal ribbon array closer to the hBN layer increases resonant absorption due to the rapid decay of the scattered fields and the resultant sharp contrast in the intensity on the top and bottom surfaces of the hBN layer. On the other hand, the reduced separation between hBN and the metal ribbons will decrease the resonant absorption of the symmetric mode, the excitation of which requires uniform field intensities along the out-of-plane direction.

These qualitative trends are well captured by our simulation, albeit with a constant scaling factor (<1) to the simulation results (see Fig. 3e caption). The decreased absorption is understandable due to known defects such as grain boundaries, and nonuniform thickness of hBN flakes as well as imperfections in metal nanostructures. The defects in hBN impact the antisymmetric mode more severely due to the confined fields within hBN (see Supplementary Fig. 3 for power distribution of image polaritons). From spectra measured from devices with different $p$, the in-plane momenta are extracted using the approximate equation $k = 2\pi m/p$, where $m$ is the order of resonance and the effective indices are obtained by $n_{\text{eff}} = k/k_0$, with $k_0$ being the free-space wavevector (Fig. 3f) (for all the measured spectra, see Supplementary Figs. 4 and 5). The experimental results agree well with the theoretical dispersions and show two distinct branches, namely the symmetric and antisymmetric modes. For both modes, the in-plane momentum increases as the gap size decreases, though the antisymmetric mode is less sensitive. Our strategy of using image polaritons improves the polariton confinement, producing record-high $n_{\text{eff}}$ of 132 and 85 for the antisymmetric and symmetric modes, respectively, the former of which is among the highest values reported for all polariton modes supported in polar dielectrics such as hBN, $MoO_3$, and SiC.

**Naturally abundant hBN vs. isotopically enriched h¹⁰BN.** The utilization of hBN enriched with $^{10}\text{B}$ isotopes made possible the observation of high-momenta HiPP modes with inherently low spectral weights. The theoretical polariton dispersions for naturally abundant hBN as well as h¹⁰BN are provided in Fig. 4a. These dispersions coincide after the offset between the Reststrahlen bands between the two forms is compensated by a $34 \, \text{cm}^{-1}$ spectral shift of the naturally abundant material towards higher frequencies. In contrast to the case of acoustic plasmons in graphene where the dispersion is linear due to the square-root relation between momentum and dielectric function, the dispersion of image polaritons in hBN is nonlinear due to the distinct dielectric function dominated by strong infrared absorption of transverse optical phonons. It should be emphasized that despite the different forms of the dispersions, the concept of additional field confinement due to the proximal

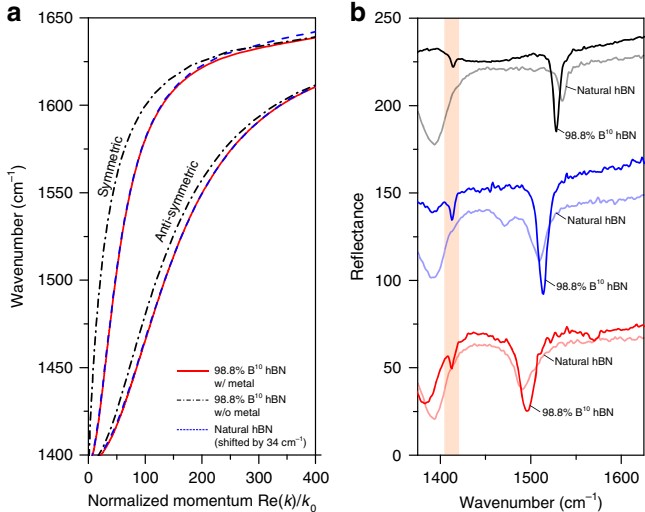

**Fig. 4 Naturally abundant hBN vs. ¹⁰B-enriched h¹⁰BN. a** The dispersion of hyperbolic image polaritons for the naturally abundant (blue dashed line) and isotopically enriched h¹⁰BN (red solid lines) for the gap size of 3 nm. The black alternating lines represent the dispersions of normal HPhPs without a mirror in isotopically enriched hBN. **b** The measured reflection spectra for the two hBN for different gap sizes of 3, 8, and 20 nm given the periodicity of 150 nm. In all cases, the spectra from naturally abundant hBN cases are shifted toward higher frequencies by $34 \, \text{cm}^{-1}$ to align the Reststrahlen bands for better visibility. The orange shaded region shows the presence of the resonances from the anti-symmetric modes, which can be observed only in the h¹⁰BN cases.

metal and image polaritons remains valid. The far-field spectra measured from the resonators using the natural hBN and h¹⁰BN with nominally identical geometric dimensions are provided in Fig. 4b. After compensating for the offset, the resonances for the symmetric mode measured from the naturally abundant hBN devices with different gap sizes are well aligned to those of the h¹⁰BN structures, consistent with Fig. 3a. The corresponding $Q$-factors are 60, 75, and 155 for $g = 3$, 8, and 20 nm, respectively. For the h¹⁰BN case, much larger $Q$'s of 81, 137, and 209 were measured. For all the gap sizes, the resonant absorption of the symmetric mode for the h¹⁰BN case is much higher than the resonant absorption with natural hBN due to the larger internal resonance enhancement given as $|1 - t_{12}t_{21}\exp(-2\pi/(\text{FoM}))|^{-2}$, where $t_{12}$ and $t_{21}$ represent the modal transmission coefficients of the HPhPs from the metal-free region to the metal-coupled region and vice versa, respectively (see Supplementary Note 2). Due to the larger $t_{12}$ and $t_{21}$ for the anti-symmetric mode (see Supplementary Fig. 6), the internal resonance enhancement factor for the h¹⁰BN can be an order of magnitude larger than naturally abundant hBN. The larger internal resonance enhancement makes resonances from the antisymmetric mode in h¹⁰BN clearly observable, which are difficult to resolve for naturally abundant hBN due to the small spectral weight.

**Loss mechanisms for hyperbolic image phonon polaritons in hBN.** Due to the zigzag propagation of HPhPs in an hBN slab resulting from the restricted propagation angle within the material, surface scattering is expected to play an important role on the damping of the HiPPs. The rate of the surface scattering events increases with the hyperbolic ray propagation angle of the HPhPs $\theta_{\text{ph}} = \arctan\left(\frac{\sqrt{\varepsilon_y(\omega)}}{i\sqrt{\varepsilon_t(\omega)}}\right)$ measured from the optical axis[10] as shown in Fig. 5a. Based on this concept, we have developed an analytical

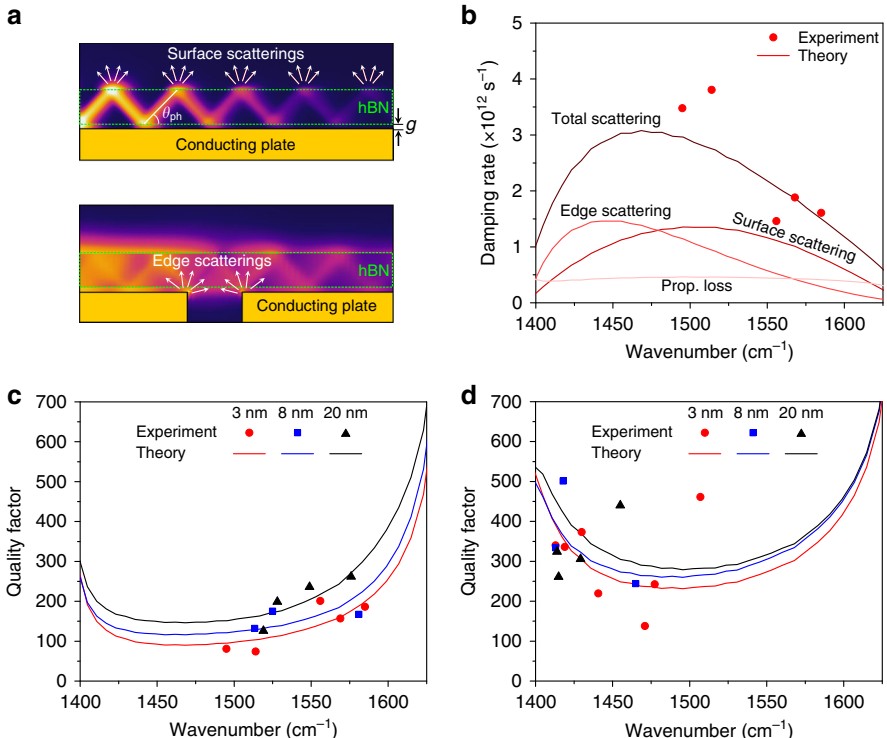

**Fig. 5 Loss mechanism for hyperbolic image polaritons in h¹⁰BN. a** Schematic illustration for the surface scatterings mediated by the hyperbolic rays inside hBN. **b** The loss contributions from three different damping mechanisms calculated using the analytical model in Eq. (1) for $g = 3$ nm. The red circles are the total scattering rates measured from the samples with different periodicities. The experimental (symbols) and analytical results (solid lines) for the quality factors obtained from the resonances for **c** the symmetric mode and **d** antisymmetric mode.

model for the scattering rate $\gamma_{\text{total}}$ of the image polariton based on the formulas for conventional Fabry-Perot resonators[52], given as

$$\gamma_{\text{total}} = \gamma_e + \gamma_p + \gamma_s = \frac{-\ln(T_{12}T_{21})}{\tau_p} + 2|\text{Im}(k)|v_g$$
$$+ \frac{v_g|\tan(\theta_{\text{ph}})|}{t}\frac{A}{1 + \left(\frac{g}{g_c}\right)}, \quad (1)$$

where $\gamma_e$, $\gamma_p$, and $\gamma_s$ represent the scattering rates for three major damping pathways considered in this model; the scatterings at the interface between the resonator units, propagation loss, and hyperbolic surface scattering (see Supplementary Note 2). $T_{12}$ and $T_{21}$ represent the modal transmittances of the image polariton through the resonator units given as $|t_{12}|^2$ and $|t_{21}|^2$, respectively. $\tau_p$ and $v_g$ denote the dwell time within a resonator unit and the group velocity of the phonon polariton, respectively. $A$ is a dimensionless fitting parameter that represents the intensity of the scattering, while $g_c$ is a fitting parameter representing a critical gap size, beyond which the dispersion of an image polariton becomes less sensitive to the gap size. The surface scattering rate is inversely proportional to the gap size based on the fact that the hyperbolic rays become more evident as the gap size decreases. The best fit to the experimental results is obtained when $A = 0.118$ and $g_c = 21.03$ nm. The total damping rates estimated from our analytical model for $g = 3$ nm agree well with the experimental results as shown in Fig. 5b (see the antisymmetric case in Supplementary Fig. 7). Due to the long lifetime of the HPhPs in h¹⁰BN, the contribution from the propagation loss is insignificant. The surface scattering is most severe at frequencies near the middle of the Reststrahlen band. The decrease in the contribution from the surface scattering near $\omega_{\text{TO}}$ is attributed to small $\theta_{\text{ph}}$, while in the spectral vicinity of $\omega_{\text{LO}}$, this is due to small $v_g$. Edge scattering plays an important role at around $\omega_{\text{TO}}$, where $T_{12}$ and

$T_{21}$ become small due to the smaller effective index of the image polariton.

Our model describes the trend of the $Q$ as a function of the device gap and other physical parameters with good accuracy. The $Q$'s extracted from the resonances for the symmetric mode agree well with the theoretical results calculated using $\omega/\gamma_{\text{total}}$ (Fig. 5c), supporting our conjecture that the total scattering rate increases with decreasing the gap size (see Supplementary Fig. 8 for the extraction method for the $Q$). The increase in the $Q$-factor at higher frequencies is also explained with the theory that associates the lower damping with decreasing $v_g$. Our theoretical model predicts that the $Q$'s for the antisymmetric mode are larger than the $Q$'s for the symmetric mode due to the smaller group velocity and larger modal transmittances. Indeed, the $Q$'s measured for the antisymmetric mode are much larger than the $Q$'s for the symmetric mode (Fig. 5d). The fit with the experimental results is best when $A = 0.108$ and $g_c = \infty$, which shows that the antisymmetric mode is less sensitive to the gap size. In general, the decrease in the $Q$ with increasing frequency is well captured by our theory. However, the experimental results show noticeable departure from the theoretical results, which can be attributed to the sensitive nature of the antisymmetric mode to defects and material composition of hBN. Also, around 1500 cm⁻¹, where the polariton wavelength of the antisymmetric mode becomes extremely small, the edge or surface scatterings seem to be more suppressed than anticipated from the theory. The numerical results also predict such high $Q$'s around 1500 cm⁻¹ (Supplementary Fig. 9). Experimentally, we observed the second-order resonance of the antisymmetric mode located at a frequency of 1507 cm⁻¹ with $n_{\text{eff}} = 132$ and $Q$ of 461. The highest $Q$ was 501 near $\omega_{\text{TO}}$, which is the among the highest value reported for any polaritonic resonators.

## Discussion

We experimentally observe strong resonances from HiPPs in $^{10}B$ isotopically enriched hBN and identified the symmetric and antisymmetric modes that simultaneously achieve ultratight field confinement and high $Q$-factors, overcoming the traditional tradeoff between these properties. Far-field observations of such ultra-confined modes become possible with our image polariton resonator that utilizes a pristine unpatterned hBN flake coupled with an array of ultraflat metallic ribbons, separated by a thin dielectric layer. The image polariton resonator features the high coupling efficiency and precise control over the gap size with nanometer accuracy, which proved essential to observe the antisymmetric mode. Most importantly, the use of unpatterned $h^{10}BN$ flakes with low optical losses allowed us to clearly observe the antisymmetric mode with extremely high $Q$'s of up to 501, which was difficult to detect with natural hBN, due to the modes low spectral weight. The symmetric modes are found to have $Q$-factors extending up to 262 and $n_{eff}$ of up to 85, yet exhibits higher coupling efficiencies to the incident field, approaching 90%. From the measured $Q$ and $n_{eff}$, we estimated that the Purcell factor could reach $1.13 \times 10^7$ for the antisymmetric mode (see Supplementary Note 3), which is an order of magnitude larger than estimated for graphene plasmons[53] and metal plasmons[54] at midinfrared frequencies. The theoretical model developed here provides a clear understanding of the loss mechanisms and a strategy to design optimal polaritonic nanoresonators. Our far-field image polariton platform and analytical model of the loss mechanism will enable researchers to harness ultra-low-loss and ultra-compressed polariton modes inaccessible via graphene or metal plasmons. This capability, in turn, will enable a series of new fundamental studies that requires ultrastrong-light–matter interactions, as well as nonlinear[55,56] and nonlocal[35] effects. Also, the ability of our platform to efficiently couple to far-field light will be essential for the development of high-contrast polaritonic sensors, optoelectronic devices, and light-emitting devices. While plasmons in graphene or noble metals have been considered the most promising route to extreme subwavelength confinement, our findings show that image phonon polaritons in hBN can outperform plasmons for a broad variety of applications.

## Methods

**hBN crystal growth**. The monoisotopic hBN crystals were grown as described elsewhere[57]. In brief, 4.0 wt% of boron which was 99.22% the B-10 isotope was mixed with equal masses of nickel and chromium, then heated to 1550 °C to form a liquid solution. This was performed at atmospheric pressure under a nitrogen plus hydrogen gas mixture. The solution was slowly cooled which reduced the hBN solubility, causing crystals to form on the surface. These were removed from the boule with tape, after the boule reached room temperature.

**Device fabrication**. A (100) silicon wafer was heated on a hot plate at 180 °C for 5 min and treated using a standard oxygen plasma recipe for 2 min (Advanced Vacuum, Vision 320). A 60-nm-thick gold film was deposited as a sacrificial layer using an electron-beam evaporator (CHA Industries, SEC 600). Then, an alumina layer was deposited using an atomic layer deposition (ALD) system (Cambridge Nano Tech Inc., Savannah). On top of the alumina film, a 30-nm-thick gold film was deposited using plasma sputtering (AJA, ATC 2200). The metal film was patterned by using an electron-beam lithography system (Vistec, EBPG5000+) followed by an Ar$^+$ ion milling process (Intlvac, Nanoquest) at a beam current of 70 mA and an accelerating voltage of 24 V for 4 min. The remaining PMMA resist was removed by oxygen plasma treatment. The resultant metal ribbons were covered by alumina (20 nm via ALD), silicon oxide (1 nm via sputterer), amorphous silicon (530 nm via plasma-enhanced chemical vapor deposition using Plasma-Therm), titanium (1 nm via sputterer), and gold layer (60 nm via sputterer). The fabricated multilayer stack was peeled off from the silicon wafer and transferred to a glass substrate by using photocurable optical adhesive (Norland, NOA 61) to bond the two substrates. In the transferred device on the glass substrate, the order of the whole stack is inverted, meaning that the top surface of the template-stripped device was previously the bottom interface between the silicon wafer and sacrificial gold layer. The sacrificial layer was simply removed using a gold etchant (Sigma Aldrich) with high etching selectivity to the underlying alumina film. Lastly, the mechanically exfoliated hBN flakes were the atomically clean exfoliated hBN flakes with desirable thickness were identified with atomic-force microscope and subsequently transferred onto the template-stripped substrate via a standard dry-transfer method.

**Device characterization**. The reflection spectra were measured using an optical microscope coupled to a FTIR spectrometer (Vertex; Bruker) using a 36× Cassegrain objective lens. The spectral resolution was 2 cm$^{-1}$. The reflectance spectra were referenced to a gold mirror.

**Numerical simulation**. The spatial distributions of the electric fields were calculated using COMSOL Multiphysics (COMSOL Inc.).

## Data availability

The data that support the plots within this paper and other findings of this study are available from the corresponding author upon reasonable request.

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

## Acknowledgements

This research was supported by grants from the National Science Foundation (NSF) MRSEC Seed (to I.-H.L., T.L., and S.-H.O.), NSF ECCS Award #1809723 (to I.-H.L., S.-H.O., and T.L.), and the Samsung Global Research Outreach (GRO) Program (to S.-H.O.). Funding for J.D.C. and M.H. was provided by the Office of Naval Research under grant number N000141812107. X.Z., Y.L., and K.W. were supported in part by NSF DMREF Award #1922165. J.H.E. and S.L. were supported by the Materials Engineering and Processing program of the NSF (CMMI 1538127) and the II−VI Foundation for hBN crystal growth. S.-H.O. further acknowledges support from the Sanford P. Bordeau chair at the University of Minnesota. Device fabrication was performed in the Minnesota Nano Center at the University of Minnesota, which is supported by the NSF through the National Nanotechnology Coordinated Infrastructure (NNCI) under Award # ECCS-1542202. Electron microscopy measurements were performed in the Characterization Facility, which has received capital equipment from the NSF MRSEC.

## Author contributions

I.-H.L. and S.-H.O. conceived the idea. J.H.E. and S.L. synthesized h¹⁰BN crystals. I.-H.L., M.H., X.Z., and Y.L. performed device fabrication. I.-H.L. and M.H. characterized devices. I.-H.L., T.L., P.A., K.W., J.D.C., and S.-H.O. performed theoretical analysis. All authors analysed the data and wrote the paper together.

## Competing interests

The authors declare no competing interests.

## Additional information

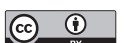

