## [Peer Review File · Nature Communications]

Reviewers' comments:

Reviewer #1 (Remarks to the Author):

The authors present an interesting device based on hybrid hBN-metal nanostructures featuring extremely confined electromagnetic modes with high quality factors of up to 500 at mid-infrared frequencies.

This is a record for nanophotonic devices at MIR frequencies. Their design goes beyond existing platforms using highly confined plasmons in graphene coupled to metallic surfaces. In their device, they used hyperbolic phonon polaritons (HPhP) in hBN coupled to their image polaritons in metallic ribbons. The phononic nature of the HPhP enables highly confined modes that are not limited by Landau damping. They further demonstrate by comparing their experimental data with their theoretical model that the losses of this new design are solely limited by surface scattering losses. This is a new and highly interesting device that could unlock potential applications at MIR frequencies such as single molecule spectroscopy and sensing applications.

Overall the paper is well written and goes straight to the point without lengthy explanations.

I recommend this paper for publication in Nature Communications.

I have added a few comments that the authors are welcome to take into account in order to improve the quality and readability of the paper.

1. In the paragraph "Gap dependence and dispersion" it was unclear to me which type of hBN you used. Is it the h10BN or the naturally abundant hBN? In the paragraph "Naturally abundant vs. isotopically enriched hBN" you analyse the differences in your reflectance measurements of naturally abundant hBN and h10BN. You clearly demonstrate that the antisymmetric mode is only visible in samples with the h10BN. I therefore suppose that the devices presented in the "Gap dependence and dispersion" are fabricated from h10BN. For clarity, I would suggest the authors to start first to present their results with different isotopically enriched hBN, show that anti-symmetric modes are only visible in h10BN and then discuss the dispersion properties of the symmetric and anti-symmetric modes in h10BN. I think that it will increase the clarity of the text.

2. The definition of the symmetric and antisymmetric modes should be sharpened. In general the definition of the antisymmetric and symmetric modes is given by the symmetry of the tangential (here in your case E_x) of the electric field in the hBN slab. In Figure 1b, it is unclear to me what component of the E field is plotted. Is it $E_x(y)$? If so, please indicate it more clearly.

3. The authors mention that the inhomogeneities and defects in the hBN film are responsible for the decreased absorption observed in the reflectance. They mention also in the method that they have measured the thickness to the hBN flakes with AFM. An AFM image in the supplementary would be useful for the readers to assess the homogeneity of the thickness throughout the hBN flakes. If you could extract an RMS of the roughness that would even be better.

4. Minor typos:

- "While these performance metrics are already impressive, their ultimate limit has yet NOT been experimentally probed and understood"

- Legend with gaps in Figure 2g.

- "...where t_{12} and t_{21} represents the modal transmission coefficients of the HPhPs" should be changed to "... where t_{12} and t_{21} represent..."

Reviewer #2 (Remarks to the Author):

1. General comments

In this manuscript, the authors achieved the highly confined polariton with high Q-factor, together with the high coupling efficiency. Not only the geometry is fully optimized with the image polariton scheme pioneered by the authors, the isotopically pure hBN sample was employed to further suppress the loss. This leads to this extremely thin layer of hBN/metasurface system absorbing ~90% of the light with extremely high Q-factor beyond $\gg 200$, which is simply stunning. I firmly believe that the manuscript merits the publication in Nature Communications.

However, before further consideration for publication, I would like the authors to address the following technical comments below, and I would like to review the manuscript again with the authors' updates.

2. Major comments

(1) The role of the periodic metallic ribbons

As the conduction plane, the authors use the periodic metallic ribbons with some periodicity (p), while some of the simulations (e.g. Fig. S1) appear to have been done assuming an extended conducting plane. Within the manuscript, the role of the periodic metal array was indeed unclear. Do I correctly understand that the periodicity is essentially adding a kick to the in-plane momentum as typically done with a grating to launch a surface plasmon polariton (SPP)? The authors should clarify the role of this somewhere early in the manuscript, as well as mentioning at the SI note 1 how does the (lateral) gap in the conducting plane potentially affect the dispersion curve.

(2) The estimation of the Purcell factor

At conclusion section, the authors estimate the Purcell factor of the polariton-based resonator to be exceeding 107, based on the quality factor and effective refractive index for the symmetric mode. However, the Purcell factor also depends on the mode volume – the highly confined nature of the polariton mode was discussed throughout the manuscript, while the quantitative value of the mode volume was never discussed, as long as I am aware of. As the gigantic Purcell factor is one of the signatures of this resonator, the authors should clarify the quantification of the mode volume.

Also, in the end it was unclear whether the symmetric mode or the anti-symmetric mode is more beneficial for any further applications. At least for the Purcell factor, the anti-symmetric mode seems to be more advantageous, as it has higher Q-factor as well as mode-confinement (see my comment below). One of the unique natures of the use of h10BN is also in that the anti-symmetric mode is observable. Letting aside the lower coupling efficiency for this mode, the authors should compute the Purcell factor for the anti-symmetric mode as well.

(3) Confinement of the anti-symmetric mode

I found the authors' claim on the confinement and gap-size dependence of the anti-symmetric mode to be confusing. The authors claim at the early stage of the manuscript that the mode confinement for the AS mode is stronger than the S mode based on the cartoon in Fig. 1c (p.3: Due to the out-of-phase charge oscillation,... leading to stronger polariton confinement than for the symmetric mode). If the field is confined at the gap between the hBN and conducting plane, I would expect the momentum, Q-factor, loss etc. to be highly dependent on the gap.

However, in contrary, the authors find that the reflection spectra and loss mechanism to be insensitive to the gap, which is against the intuition. (p.4: Such spatial distributions also explain why the anti-symmetric mode is less sensitive to the gap size; p.6: ... which shows the anti-symmetric mode is less sensitive to the gap size). At pg. 4, the authors attempt to give some explanations based on Fig. 2f, merely based on which the mechanism of the insensitivity is honestly unclear. If the calculations in Fig. 2e and 2f are repeated for another gap and demonstrate how the field profiles are altered, it may be more convincing.

The authors should strengthen the discussion on this aspect.

3. Minor comments

(1) The role of objective lens

The authors use FTIR microscope. Clearly this is necessary to observe the region where hBN/metrasurface is formed, but is this also important to launch the polariton by adding some in-plane momentum?

(2) h10BN

In Figure 1 and 2, the authors should clearly indicate they measure h10BN, not the natural one. I was confused for a while on which sample the authors are measuring.

(3) Fig. 1a

Is this mainly for the symmetric mode? Do I correctly understand the profile is dominated by the symmetric mode because the symmetric mode has less momentum mismatch? If you plot the field amplitude with log scale for example, can you also see the anti-symmetric mode?

(4) Fig. 2g

The y-axis should be scaled from 0 – 100%.

(5) Fig. 3

The labels are missing. In Fig. 3a, I believe the left one is for the symmetric mode and the right one is for the anti-symmetric mode. If so, the curves need to be clearly labeled and clarified in the caption.

The three curves, I believe, correspond to $g = 20$ nm, 8 nm, 3 nm, from top to bottom. Again, this needs to be clearly labeled in the figure.

(6) Introduction

For the general readership of Nature Communications, it may help to clarify the potential broader applications of phonon/plasmon polaritons at the very beginning of the introduction with some citations, e.g. from Oh, Basov, and Hillenbrand group etc.

Reviewer #3 (Remarks to the Author):

“Pushing the Polariton Confinement Limits with Low Losses using Image Polaritons in Boron Nitride” by Li et al. demonstrates highly confined, low-loss modes associated with hexagonal boron nitride near metallic gratings, which serve not only to couple far-field light to the polaritons, but also change their dispersion when the gap is very small. By performing scattering experiments, the authors conclude to have observed both the symmetric lowest-order metal-dielectric-hBN mode, as well as the antisymmetric mode, the latter of which was concluded to have an effective index of 130 and a quality factor of roughly 500. Their results were in part assisted by: not patterning the BN directly, and using isotopically enriched BN, similar to a previous work by some of the authors in Nature Materials.

In general, the results seem correct and the platform is new (in the context of phonon polaritons) and may enable advances in light-matter interactions as well as nanophotonics with phonon polaritonics. In my opinion, the one of the strongest points is the simultaneous figures of merit achieved (confinement and loss).

However, the manuscript needs to be clarified further. In particular:

1. The authors compare their system to a somewhat similar system in graphene (Science 360.6386 (2018): 291-295.). I would like to see a more in-depth discussion about the similarities and differences between the authors' system and this one. In particular, two questions I have are:

1a. In the graphene system, the dispersion is fundamentally altered by the metal (becoming linear), while according to Fig. 2i of the current manuscript (and Fig. 1b of the supplement), the dispersion is qualitatively similar, and simply pushed to larger momenta for smaller gaps. Could the authors comment on why their system is so different?

1b. Could the authors also comment on what transverse confinement (e.g., the mode length) is in their system, as a function of gap size? How much of the energy resides in the gap? How does this compare to the analogous system in graphene?

2. What is the origin of the critical gap parameter in the surface scattering model?

3. Comparing the theoretical damping rate to the measured damping rates of the polaritons indicates considerable discrepancies. The authors attribute this to the sensitive nature of the antisymmetric mode scattering to defects and other imperfections. Could the authors back this attribution up through some (simple) simulations including roughness or other defects (simply to show sensitivity of the quality factor to realistic roughness, rather than to quantitatively account for the observations). Such considerations could also in principle determine how much variation in the quality factor is to be expected. Further complicating the authors claim is that even for the symmetric case, the larger gap sizes (8 and 20 nm), the trend in the measurements seems to not follow the predictions either. Generally, I would like to see the discussion of possible discrepancies expanded upon.

4. Exactly which field is being plotted in Fig. 1a?

5. Finally, some of the points made in the introduction/abstract need to be qualified, as they can potentially confuse a reader newer to the field.

5a. For example, "Polaritons in two-dimensional (2D) materials provide extreme light confinement that is not possible with conventional metal plasmonics" needs to be toned-down, as metal nanogap structures enable much tighter confinement (though of course, in hBN, the losses are much better, and the spectral range where confinement happens is different).

5b. Furthermore, some statements, as written, appear to not be correct. For example: "Recent works on PhPs in 2D materials such as hBN have reported effective indices of a few tens - several times larger than values previously reported for plasmons in metallic nanostructures or graphene". In graphene, effective indices of >100 are somewhat commonplace (as just two examples, take Nature materials 14.4 (2015): 421-425 or Nature 557.7706 (2018): 530-533).

Response to Reviewers – NCOMMS-20-03356-T

Reviewer #1 (Remarks to the Author):

“The authors present an interesting device based on hybrid hBN-metal nanostructures featuring extremely confined electromagnetic modes with high quality factors of up to 500 at mid-infrared frequencies.

This is a record for nanophotonic devices at MIR frequencies. Their design goes beyond existing platforms using highly confined plasmons in graphene coupled to metallic surfaces. In their device, they used hyperbolic phonon polaritons (HPhP) in hBN coupled to their image polaritons in metallic ribbons. The phononic nature of the HPhP enables highly confined modes that are not limited by Landau damping. They further demonstrate by comparing their experimental data with their theoretical model that the losses of this new design are solely limited by surface scattering losses.

This is a new and highly interesting device that could unlock potential applications at MIR frequencies such as single molecule spectroscopy and sensing applications.

Overall the paper is well written and goes straight to the point without lengthy explanations.

I recommend this paper for publication in Nature Communications.

I have added a few comments that the authors are welcome to take into account in order to improve the quality and readability of the paper.

1. In the paragraph “Gap dependence and dispersion” it was unclear to me which type of hBN you used. Is it the h10BN or the naturally abundant hBN? In the paragraph “Naturally abundant vs. isotopically enriched hBN” you analyse the differences in your reflectance measurements of naturally abundant hBN and h10BN. You clearly demonstrate that the antisymmetric mode is only visible in samples with the h10BN. I therefore suppose that the devices presented in the “Gap dependence and dispersion” are fabricated from h10BN. For clarity, I would suggest the authors to start first to present their results with different isotopically enriched hBN, show that anti-symmetric modes are only visible in h10BN and then discuss the dispersion properties of the symmetric and anti-symmetric modes in h10BN. I think that it will increase the clarity of the text.”

Response: We followed Reviewer 1’s suggestions and clarified which type of hBN was used in the main manuscript.

“2. The definition of the symmetric and antisymmetric modes should be sharpened. In general the definition of the antisymmetric and symmetric modes is given by the symmetry of the tangential (here in your case E_x) of the electric field in the hBN slab. In Figure 1b, it is unclear to me what component of the E field is plotted. Is it $E_x(y)$? If so, please indicate it more clearly.”

Response: Figs. 1b and 1c were not numerical results but schematic diagrams illustrating how fields will form depending on the symmetry of a mode. In the revised version, we have updated Figs. 1b and 1c with numerically calculated results. We have plotted normalized total electric fields instead of x or y component to illustrate optical power distribution in the gap vs. hBN region depending on the mode symmetry, which will determine the characteristics of a mode. We have also included the numerical results for the tangential components of the symmetric and anti-symmetric mode for clarity in the supplementary figure S1.

“3. The authors mention that the inhomogeneities and defects in the hBN film are responsible for the decreased absorption observed in the reflectance. They mention also in the method that they have measured the thickness to the hBN flakes with AFM. An AFM image in the supplementary would be useful for the readers to assess the homogeneity of the thickness throughout the hBN flakes. If you could extract an RMS of the roughness that would even be better.”

Response: We have added an AFM image of the h¹⁰BN flake used in our experiment (supplementary figure S2), which shows an atomically smooth surface of the flake. Also, we provide the RMS value so that readers can assess the homogeneity of a flake.

“4. Minor typos:

- “While these performance metrics are already impressive, their ultimate limit has yet NOT been experimentally probed and understood”
- Legend with gaps in Figure 2g.
- “...where t_{12} and t_{12} represents the modal transmission coefficients of the HPhPs” should be changed to “... where t_{12} and t_{21} represent...”

Response: Thank you and we have corrected them in the revised version.

Reviewer #2 (Remarks to the Author):

“1. General comments

In this manuscript, the authors achieved the highly confined polariton with high Q-factor, together with the high coupling efficiency. Not only the geometry is fully optimized with the image polariton scheme pioneered by the authors, the isotopically pure hBN sample was employed to further suppress the loss. This leads to this extremely thin layer of hBN/metasurface system absorbing ~90% of the light with extremely high Q-factor beyond >>200, which is simply stunning. I firmly believe that the manuscript merits the publication in Nature Communications. However, before further consideration for publication, I would like the authors to address the following technical comments below, and I would like to review the manuscript again with the authors' updates.

2. Major comments

(1) The role of the periodic metallic ribbons

As the conduction plane, the authors use the periodic metallic ribbons with some periodicity (p), while some of the simulations (e.g. Fig. S1) appear to have been done assuming an extended conducting plane. Within the manuscript, the role of the periodic metal array was indeed unclear. Do I correctly understand that the periodicity is essentially adding a kick to the in-plane momentum as typically done with a grating to launch a surface plasmon polariton (SPP)? The authors should clarify the role of this somewhere early in the manuscript, as well as mentioning at the SI note 1 how does the (lateral) gap in the conducting plane potentially affect the dispersion curve.”

Response: As the Reviewer 2 mentioned, the periodicity supplements the in-plane momentum required to satisfy the momentum mismatch between free-space light and polaritons. In the revised version, we have added two sentences on p. 4 to describe the role of the metal ribbon array and one reference to direct readers for more details.

“(2) The estimation of the Purcell factor

At conclusion section, the authors estimate the Purcell factor of the polariton-based resonator to be exceeding 107, based on the quality factor and effective refractive index for the symmetric mode. However, the Purcell factor also depends on the mode volume – the highly confined nature of the polariton mode was discussed throughout the manuscript, while the quantitative value of the mode volume was never discussed, as long as I am aware of. As the gigantic Purcell factor is one of the signatures of this resonator, the authors should clarify the quantification of the mode volume.

Also, in the end it was unclear whether the symmetric mode or the anti-symmetric mode is more beneficial for any further applications. At least for the Purcell factor, the anti-symmetric mode seems to be more advantageous, as it has higher Q -factor as well as mode-confinement (see my comment below). One of the unique natures of the use of h10BN is also in that the anti-symmetric mode is observable. Letting aside the lower coupling efficiency for this mode, the authors should compute the Purcell factor for the anti-symmetric mode as well.”

Response: We agree with Reviewer 2 that the anti-symmetric mode can be advantageous. That’s why we calculated the Purcell factor for this mode, which seems to have a larger value because of tighter polariton confinement and higher Q -factor. In our case, the resonator is not confined along the long axes of the ribbons, which makes it difficult to define a mode volume. Thus, we estimate Purcell factor assuming that a resonator with square patch pattern instead of ribbon will have similar optical responses. In the revised version, we have included details on how we estimated the Purcell factor based on this assumption as follows:

“Supplementary note 3. Purcell factor estimation

To calculate Purcell factor, mode volume should be first determined. Since near-fields of image polaritons are indefinitely continuous along the long axes of ribbons in our case, however, the mode volume of image polaritons is in principle also indefinite. Instead, we approximately estimate the mode volume V_{eff} with $V_{\text{eff}} = (\text{periodicity})^2 \times (\text{gap} + \text{hBN thickness} + \text{evanescent field decay length})$ assuming that an image polariton resonator with square patches would have similar polariton properties. Then, the Purcell factor F_{P} can be calculated from

$$F_{\text{P}} = \frac{3}{4\pi^2} \left(\frac{\lambda_0}{n}\right)^3 \left(\frac{Q}{V_{\text{eff}}}\right), \quad (9)$$

where λ_0 , n , and Q represent a free-space wavelength of incident light, a refractive index of a cavity, and a quality factor of a resonance.

The Purcell factor of 1.13×10^7 , which was mentioned in the main article, was calculated from one of the anti-symmetric resonances with the largest effective index of 132 and the quality factor of 461 at a wavenumber of 1507 cm^{-1} for which the periodicity of the metal ribbon array is 100 nm and the gap size is 3 nm. Thus, the mode volume is calculated as $(100 \text{ nm})^2 \times (3 \text{ nm} + 23 \text{ nm} + 7 \text{ nm})$, with the estimated decay length of the evanescent fields into the air being 7 nm. For the refractive index of a cavity n , the refractive index of alumina at 1507 cm^{-1} is used as an approximation.”

“(3) Confinement of the anti-symmetric mode

I found the authors’ claim on the confinement and gap-size dependence of the anti-symmetric mode to be confusing. The authors claim at the early stage of the manuscript that the mode confinement for the AS mode is stronger than the S mode based on the cartoon in Fig. 1c (p.3: Due to the out-of-phase charge oscillation, ... leading to stronger polariton confinement than for

the symmetric mode). If the field is confined at the gap between the hBN and conducting plane, I would expect the momentum, Q-factor, loss etc. to be highly dependent on the gap. However, in contrary, the authors find that the reflection spectra and loss mechanism to be insensitive to the gap, which is against the intuition. (p.4: Such spatial distributions also explain why the anti-symmetric mode is less sensitive to the gap size; p.6: ... which shows the anti-symmetric mode is less sensitive to the gap size). At pg. 4, the authors attempt to give some explanations based on Fig. 2f, merely based on which the mechanism of the insensitivity is honestly unclear. If the calculations in Fig. 2e and 2f are repeated for another gap and demonstrate how the field profiles are altered, it may be more convincing. The authors should strengthen the discussion on this aspect.”

Response: As we described in Fig. 1, the anti-symmetric modes consists of oscillations of phonon polaritons with different polarities on the bottom and top of hBN flake, which in turn, confine most of light within the hBN flake rather than the gap region. This contrasts with the symmetric case where light is mostly concentrated within the gap region due to the polarities of charge being the same on opposite sides of the hBN flake. In the revised manuscript, we added Fig. S3 where we show how the energy of the symmetric and anti-symmetric mode distributes over the gap and hBN region. Also, the numerical results for the field profile of the symmetric and anti-symmetric modes have been added as the new Figs. 1b and c will help clarify this issue.

“3. Minor comments

(1) The role of objective lens

The authors use FTIR microscope. Clearly this is necessary to observe the region where hBN/metasurface is formed, but is this also important to launch the polariton by adding some in-plane momentum?”

Response: The in-plane momentum provided by an objective lens is much smaller than what required to excite phonon polaritons. As stated above, the momentum mismatch is overcome through the presence of the underlying metallic grating. While the 36x FTIR objective used for these studies provides off-normal excitation with a weighted average incident angle on the order of 25°. This same objective had been previously used to study the role of the angle of incidence on the reflection of hBN flakes in the supplemental information of a previous work (J. D. Caldwell et al. *Nat. Commun.* **5**, 5221 (2014)). In that study, while the incident angle is shown to have an influence on the reflection spectra, as the ability to couple to the ordinary and extraordinary portions of the dielectric function is changed, in no case were polaritonic modes observed from these flakes (they were observed from nanostructures in the main text, but in that case the momentum mismatch is overcome from the structuring, not the objective).

Comment: “(2) h10BN

In Figure 1 and 2, the authors should clearly indicate they measure h10BN, not the natural one. I was confused for a while on which sample the authors are measuring.”

Response: We have clarified this issue in the revised version.

Comment: “(3) Fig. 1a

Is this mainly for the symmetric mode? Do I correctly understand the profile is dominated by the symmetric mode because the symmetric mode has less momentum mismatch? If you plot the field amplitude with log scale for example, can you also see the anti-symmetric mode?”

Response: The field profile in Fig. 1a has been numerically calculated by using a dipole source. Therefore, not only symmetric and anti-symmetric mode but also all the higher order modes contribute to the field profile. So it is difficult to resolve the anti-symmetric mode even if the field profile is plotted with log scale. In order to resolve the anti-symmetric mode, the field profile should be Fourier-transformed.

“(4) Fig. 2g The y-axis should be scaled from 0 – 100%.

(5) Fig. 3 The labels are missing. In Fig. 3a, I believe the left one is for the symmetric mode and the right one is for the anti-symmetric mode. If so, the curves need to be clearly labeled and clarified in the caption.

The three curves, I believe, correspond to $g = 20$ nm, 8 nm, 3 nm, from top to bottom. Again, this needs to be clearly labeled in the figure.”

Response: We have addressed all of these issues in the revised version.

Comment: “(6) Introduction

For the general readership of Nature Communications, it may help to clarify the potential broader applications of phonon/plasmon polaritons at the very beginning of the introduction with some citations, e.g. from Oh, Basov, and Hillenbrand group etc.”

Response: We have added a sentence to describe potential applications in the revised version.

Reviewer #3 (Remarks to the Author):

“Pushing the Polariton Confinement Limits with Low Losses using Image Polaritons in Boron Nitride” by Li et al. demonstrates highly confined, low-loss modes associated with hexagonal boron nitride near metallic gratings, which serve not only to couple far-field light to the polaritons, but also change their dispersion when the gap is very small. By performing scattering experiments, the authors conclude to have observed both the symmetric lowest-order metal-dielectric-hBN mode, as well as the antisymmetric mode, the latter of which was concluded to have an effective index of 130 and a quality factor of roughly 500. Their results were in part assisted by: not patterning the BN directly, and using isotopically enriched BN, similar to a previous work by some of the authors in Nature Materials.

In general, the results seem correct and the platform is new (in the context of phonon polaritons) and may enable advances in light-matter interactions as well as nanophotonics with phonon polaritonics. In my opinion, the one of the strongest points is the simultaneous figures of merit achieved (confinement and loss).

However, the manuscript needs to be clarified further. In particular:

1. The authors compare their system to a somewhat similar system in graphene (Science 360.6386 (2018): 291-295.). I would like to see a more in-depth discussion about the similarities

and differences between the authors' system and this one. In particular, two questions I have are:

1a. In the graphene system, the dispersion is fundamentally altered by the metal (becoming linear), while according to Fig. 2i of the current manuscript (and Fig. 1b of the supplement), the dispersion is qualitatively similar, and simply pushed to larger momenta for smaller gaps. Could the authors comment on why their system is so different?"

Response: We thank Reviewer 3 for giving us an opportunity to clarify this important point. We agree that it is important to explain the difference between image polaritons in graphene vs. hBN. In the revised manuscript, we added a new paragraph in the sub-section 'Image polaritons: concept and dispersion characteristics' to explain this critical issue (copied below):

"The dispersion relation for 2D image polaritons depends on the specific origin of the polariton modes i.e. plasmon-, phonon- or exciton-polaritons. In the case of 2D plasmons, the dielectric response is dominated by free electrons and is given by $\varepsilon(\beta, \omega) = 1 - V_c \Pi$, where $V_c = \frac{e^2}{2\beta\epsilon_0} (1 - e^{-\beta d})$ is the Coulomb potential in Fourier representation and $\Pi = \frac{D\beta^2}{\omega^2 e^2}$ is the electron polarizability. Both the Coulomb and polarizability term are universal for *all* 2D electron gas, and d is the separation between the metal with the 2D material, where the specific materials information is encoded in the Drude weight D . The plasmon dispersion can be obtained by setting $\varepsilon(\beta, \omega) = 0$. In the limit $d \rightarrow 0$, the plasmon has the generic dispersion $\omega \propto \beta$, and $\omega \propto \sqrt{\beta}$ in the $d \rightarrow \infty$. Due to its linear dispersion, this underlies the origin of the use of *acoustic plasmon* for describing plasmon-polariton in proximity to metal, whose dispersion has been observed in various systems such as graphene/semiconductor in proximity to metal and graphene-insulator-graphene [35]. Phonon-polaritons in hBN, on the other hand, has a different dielectric function. Most importantly, the prerequisite for polariton (i.e. surface electromagnetic mode) across a spectral window is that $Re[\varepsilon(\omega)] < 0$. Since the infrared response of insulators are due to optical phonons, $Re[\varepsilon(\omega \sim \omega_{op})] < 0$ only in the spectral vicinity of $\omega \sim \omega_{op}$. Hence, for phonon-polaritons, the dispersion cannot be linear, and it depends on the specific form of the dielectric function. In summary, the dispersion of image polaritons in general depends on the specific origin of the polariton modes, but the concept of additional field confinement due to the proximal metal is universal."

"1b. Could the authors also comment on what transverse confinement (e.g., the mode length) is in their system, as a function of gap size? How much of the energy resides in the gap? How does this compare to the analogous system in graphene?"

Response: The transverse confinement of the polariton is generally given as $\lambda_p/2\pi$, so it can be estimated from the changes in the lateral confinement, which is given in the manuscript. We have run numerical simulations to determine how much of the energy resides in the gap depending the symmetry of a mode, the contents of which has been summarized in Fig. S3. As we already discussed, for the anti-symmetric mode, most of energy resides in hBN layers while the gap regions host most of energy for the symmetric case. The graphene case can be considered as a limiting case for the symmetric mode. In other words, the energy of graphene plasmons exclusively resides in the gap.

- We have added Fig. S3 to describe how power distributes over the gap and the hBN region depending on the symmetry of a mode.

“2. What is the origin of the critical gap parameter in the surface scattering model?”

Response: The critical gap parameters represents a gap size beyond which the loss becomes less sensitive to the gap size. We have clarified its definition in the revised version.

“3. Comparing the theoretical damping rate to the measured damping rates of the polaritons indicates considerable discrepancies. The authors attribute this to the sensitive nature of the antisymmetric mode scattering to defects and other imperfections. Could the authors back this attribution up through some (simple) simulations including roughness or other defects (simply to show sensitivity of the quality factor to realistic roughness, rather than to quantitatively account for the observations). Such considerations could also in principle determine how much variation in the quality factor is to be expected. Further complicating the authors claim is that even for the symmetric case, the larger gap sizes (8 and 20 nm), the trend in the measurements seems to not follow the predictions either. Generally, I would like to see the discussion of possible discrepancies expanded upon.”

Response: We appreciate Reviewer 3’s insightful comment. Due to the limitation on the size of exfoliated hBN, all the characterized devices has different hBN flakes, which is one of a major source for the discrepancies. The variation of thicknesses, isotope concentration among different hBN flakes and the damages during transfer processes have a definite effect on the discrepancies. Such variations over different samples can be amplified in the case of the anti-symmetric mode since the fields are mostly concentrated in hBN flakes. We have added a supplementary figure illustrating how the energies of the symmetric and anti-symmetric mode distribute over the gap and hBN region, which will give Reviewer 3 more clear idea on how the anti-symmetric mode can be more sensitive to the quality of a hBN flake.

“4. Exactly which field is being plotted in Fig. 1a?”

Response: We have plotted normalized total electric fields. In the revised version, we have mentioned which field component is being plotted clearly.

“5a. For example, “Polaritons in two-dimensional (2D) materials provide extreme light confinement that is not possible with conventional metal plasmonics” needs to be toned-down, as metal nanogap structures enable much tighter confinement (though of course, in hBN, the losses are much better, and the spectral range where confinement happens is different).

5b. Furthermore, some statements, as written, appear to not be correct. For example: “Recent works on PhPs in 2D materials such as hBN have reported effective indices of a few tens - several times larger than values previously reported for plasmons in metallic nanostructures or graphene”. In graphene, effective indices of >100 are somewhat commonplace (as just two examples, take *Nature materials* 14.4 (2015): 421-425 or *Nature* 557.7706 (2018): 530-533).”

Response: We thank Reviewer 3 for pointing out these issues. We have tone-downed and revised the sentences as the reviewer suggested. Also we have added these two references.

REVIEWERS' COMMENTS:

Reviewer #1 (Remarks to the Author):

The authors have addressed satisfactorily my comments.
I support the publication of this work in Nature Commun.

Reviewer #2 (Remarks to the Author):

My comments (as Reviewer 2) were sufficiently addressed by the authors, and I only have minor suggestions below. I in particular appreciate the new information added to the introduction, which general readership will certainly appreciate. The manuscript is ready for publication from my perspective. No further review is necessary from my end.

Fig. 3b,c - now the associated discussion on how the field is confined in the gap for the symmetric mode is really helpful to get the authors' point, together with Fig. S3. Emphasizing "Gap" (as they do in Fig. 1a) in these figures might be helpful.

pg.7, Purcell factor - in the way this is written, people would assume that the Purcell factor is calculated for symmetric mode. It may be good to emphasize this is calculated for the anti-symmetric mode.

pg.4 - the new discussion added in response to Reviewer 3 was not very clear to me, and I don't know how this is helping the discussion. In particular, it is unclear how these quantitative difference in the dispersion leads to the advantage of hBN PhP to graphene SPP. If this is not a key difference (I thought that the hBN's uniqueness arises from the damping mechanism), then it may not be worth mentioning this here, and rather move this to SI or so. Otherwise, it is important to clarify why this is important.

Also, some of the basic typos pointed out by the reviewers are not fixed yet (e.g., p.6: t12 and t12; some of the "n" for the refractive index is not italic). Please make sure these typos are taken care of before publication.

Reviewer #3 (Remarks to the Author):

I have read the authors' response to the referees: I believe the discussion of discrepancies between theory and experiment is suitable, and the relation to acoustic plasmons is now clear.

The manuscript can now be accepted for publication in Nature Communications.

Reviewer 2

Comment: *“My comments (as Reviewer 2) were sufficiently addressed by the authors, and I only have minor suggestions below. I in particular appreciate the new information added to the introduction, which general readership will certainly appreciate. The manuscript is ready for publication from my perspective. No further review is necessary from my end.”*

Response: We thank that Reviewer 2 for thoroughly reading our manuscript and helping to improve it. In the revised version, we have revised all the issues raised by the Reviewer.

Comment: *“Fig. 3b,c - now the associated discussion on how the field is confined in the gap for the symmetric mode is really helpful to get the authors' point, together with Fig. S3. Emphasizing “Gap” (as they do in Fig. 1a) in these figures might be helpful.”*

Response: We have emphasized the gap region in Fig. 3b, c with red-lined box.

Comment: *“pg.7, Purcell factor - in the way this is written, people would assume that the Purcell factor is calculated for symmetric mode. It may be good to emphasize this is calculated for the anti-symmetric mode.”*

Response: To address this comment, on page 7 we explicitly mentioned that the Purcell factor has been calculated from the anti-symmetric mode.

Comment: *“pg.4 - the new discussion added in response to Reviewer 3 was not very clear to me, and I don't know how this is helping the discussion. In particular, it is unclear how these quantitative difference in the dispersion leads to the advantage of hBN PhP to graphene SPP. If this is not a key difference (I thought that the hBN's uniqueness arises from the damping mechanism), then it may not be worth mentioning this here, and rather move this to SI or so. Otherwise, it is important to clarify why this is important.”*

Response: To address this comment, we removed the second paragraph in the subsection of “Image polaritons: Concept and dispersion characteristics” and wrote two sentences in the subsection of “Naturally abundant vs. isotopically enriched $h^{10}\text{BN}$ ” to explain the physical origin of the difference in the dispersions. But we emphasize that despite the different dispersion forms, the strategy to bring a metal film in a close proximity to a 2D layer can be universally applied to increase the polariton confinement of any polariton-supporting 2D material.

Comments: *“Also, some of the basic typos pointed out by the reviewers are not fixed yet (e.g., p.6: t_{12} and t_{12} ; some of the “n” for the refractive index is not italic). Please make sure these typos are taken care of before publication.”*

Response: We have corrected ‘ t_{12} and t_{12} ’ to ‘ t_{12} and t_{21} ’. We have also italicized ‘n’.

EDITORIAL REQUESTS:

Response: We have revised our manuscript to comply with all required editorial policies.